# Validation of the Longitudinal Interval Follow-Up Evaluation for the Long-Term Measurement of Mood Symptoms in Bipolar Disorder

**DOI:** 10.3390/brainsci12121717

**Published:** 2022-12-15

**Authors:** Richard J. Porter, Will Moot, Maree L. Inder, Marie T. Crowe, Katie M. Douglas, Janet D. Carter, Christopher Frampton

**Affiliations:** 1Department of Psychological Medicine, University of Otago, Christchurch 4345, New Zealand; 2Specialist Mental Health Services, Te Whatu Ora, Christchurch 8025, New Zealand; 3School of Psychology, Speech and Hearing, University of Canterbury, Christchurch 8140, New Zealand

**Keywords:** bipolar disorder, mood rating scales, longitudinal assessment, mood assessment, psychotherapy

## Abstract

The long-term burden of symptoms is an important outcome in bipolar disorder (BD). A method which has minimal burden of assessment uses a retrospective interview, the Longitudinal Interval Follow-up Examination (LIFE), although this may be subject to problems with recall. This study examines the relationship between the retrospective LIFE scale and concurrently-rated mood rating scales in two clinical trials of 18 months of psychotherapy for patients with BD. The Montgomery-Asberg Depression Rating Scale (MADRS) and Young Mania Rating Scale (YMRS) were administered every eight to nine weeks and the LIFE was carried out every 6 months. Correlations between scores on mood rating scales and at equivalent times on the LIFE were examined, as well as of potential clinical moderators. There were significant correlations between LIFE depression ratings and concurrent MADRS score (r = 0.57) and between LIFE mania ratings and YMRS score (r = 0.40). In determining “mild depression” on the MADRS, a receiver operating characteristics (ROC) analysis showed an AUC of 0.78 for LIFE scores. Correlations, particularly for depression scores, were high even when the LIFE rating was several months before the interview, suggesting that the LIFE has validity in examining the burden of mood symptoms over time, with relatively little burden of assessment. Future research should examine the relationship between symptom burden and quality of life measured in this way.

## 1. Introduction

Mood disorders (bipolar disorder [BD] and major depressive disorder [MDD]) are long-term, involve frequent relapses, and carry a significant burden of ongoing mood symptoms. Diagnostically, MDD is defined by the presence of episode(s) of major depression, and BD is defined by episode(s) of mania (BD I) or episode(s) of hypomania and major depression (BD-II) [1]. Studies have suggested that patients with mood disorders spend a considerable amount of time in subsyndromal mood states, which are distressing and damaging to their functioning and relationships [2,3,4,5]. This has led to attempts to measure the “burden of mood symptoms” over longer periods of time. However, monitoring symptoms long-term is challenging. It is burdensome for patients to complete repeated clinician administered mood rating scales, and self-administered rating scales may be completed only by a small proportion of patients even when done online [6]. While methods of monitoring mood are being investigated which involve the use of smart phones, actigraphs and other wearable devices (Ecological Momentary Assessment—EMA) [7], the relationship between these new technologies and mood as rated by clinician-administered rating scales is not yet clear [8]. Furthermore, EMA involves a significant time burden for patients (for example 6.5 h total, on average, over 145 days [9]) and some patients may find this intrusive.

Various methods of prospectively assessing mood symptoms have been developed, perhaps most notably the NIMH life charting method which uses detailed monthly interviews. This method has been widely used and is well validated, including in research studies [10,11,12]. This life charting method has both clinician and patient versions, which can be used depending on the frequency of clinician contact. However, even monthly interviews may be unduly burdensome in some health care settings or in pragmatic clinical trials, which attempt to minimize the burden of assessment. It is important that in large, pragmatic clinical trials in psychiatry, outcomes need to be easy to measure [13] and it is possible that useful, and in some cases more complete data can be obtained by less frequent interviews.

The Longitudinal Interval Follow-up Evaluation (LIFE) was developed to assess the longitudinal course of psychiatric disorders retrospectively and at relatively lengthy intervals, in order to allow clinicians and researchers to assess the dates of individual episodes of a disorder and to rate the overall burden of symptoms [14]. Weekly data are elicited retrospectively, in interview format, usually at 6-month intervals. The method allows for the overall quantification of mood burden, both that related to depression, mania and total, i.e., depression and mania summed. By determining whether syndromal symptoms last 2 weeks, relapse into a depressive episode or a manic episode can be determined. This method has been used in a number of longer-term studies in bipolar disorder (BD) to determine time to relapse and overall mood morbidity [15,16]. The LIFE was designed to be used in a systematic way and therefore to collect data across centres which would be equivalent and could potentially be compared or pooled. However, given the retrospective nature of the LIFE, there are concerns that participants’ recollections may be inaccurate, particularly at the beginning of the time-frame being discussed. It may also be unduly influenced by the mood state at the time of undertaking the LIFE interview—and therefore fail to give a good account of the whole time period. Furthermore, it gives an impression of mood state without a detailed quantification of symptoms, as would concurrent mood rating scales. Finally, it is a wholly subjective measure of mood symptoms.

In two clinical trials of Interpersonal and Social Rhythm Therapy (IPSRT) for BD, we have used the retrospective LIFE interview [14] to assess the long-term (over 18 months) outcome of this psychotherapy when added to pharmacotherapy—with different comparison therapies in each study. In one study, the primary outcome measure was change in total burden of mood symptoms as measured by the LIFE [16], and in the other, the time to relapse into a mood episode as measured by the LIFE [17]. In both studies, the LIFE was completed every 6 months. Furthermore, in both studies, regular (every 8–9 weeks) clinician-rated scales were conducted to measure mood symptoms; the Montgomery-Asberg Depression Rating Scale (MADRS) [18], and the Young Mania Rating Scale (YMRS) [19]. This afforded the opportunity to examine the relationship between participants’ mood ratings as retrospectively established using the LIFE interview and clinician-administered mood rating scales conducted at corresponding times (i.e., ratings for particular weeks in a six-month period elicited in the LIFE could be compared with mood rating scales actually completed in that week).

### Aims

The aim of the analysis described here is therefore to examine the following questions using data from these two studies:What is the relationship of the retrospective LIFE rating at a particular time-point to the MADRS or YMRS score completed at that time-point?When compared at the same time-point, how well do categorical determinations of whether a patient is ‘in episode’ determined by the LIFE accord with that determination made by a concurrent MADRS or YMRS?What factors moderate the relationship between these measures?

## 2. Methods

Data are from patients in two randomized controlled trials (RCT) of IPSRT for BD, referred to as Study 1 [16] and Study 2 [17].

### 2.1. Inclusion/Exclusion Criteria

#### 2.1.1. Study 1

This study recruited adolescents and young adults (aged 15–36 years) with BD-I, BD-II, and BD not otherwise specified. Participants were recruited from a range of services in Canterbury, New Zealand, including mental health services and general practitioners. Because Study 2 included only BD-I or BD-II, the four patients in Study 1 with BD-NOS were excluded from the analysis. There were no criteria regarding mood state at entry.

#### 2.1.2. Study 2

This study recruited adults (18–64 years) with a diagnosis of BD-I or BD-II, and who did not meet the criteria for an episode of depression, mania, or mixed state at baseline. All patients had been discharged from a publicly-funded mental health service in Canterbury, New Zealand, within the previous 3 months.

Exclusion criteria for both studies were a primary diagnosis of schizophrenia, schizoaffective disorder, or severe substance use disorder.

### 2.2. Interventions

In both studies, IPSRT was delivered according to a manualized protocol. Details are presented in Inder et al. [16] and Crowe et al. [17].

In Study 1, patients were randomized to receive IPSRT or Specialist Supportive Care (SSC). In Study 2, patients were randomized to IPSRT or TAU. Patients randomly assigned to the TAU remained under care from their general practice physician and did not receive psychotherapy, therefore they did not have regular clinical rating scales.

### 2.3. Measures

The Structured Clinical Interview for DSM–IV Axis I Disorders (SCID-I) [20] was used to confirm psychiatric diagnoses by a research nurse.

The cumulative burden of mood symptoms was measured using the LIFE [14], retrospectively every 6 months, commencing from baseline. Patients were rated on a 1–6 scale, where 1—no symptoms, 2—residual symptoms, 3—partial remission, 4—does not meet DSM criteria but has major symptoms or impairment, 5—meets definite DSM criteria for an “episode”, and 6—fulfils definite criteria for an “episode” with the presence of either psychotic symptoms or extreme impairment in functioning. Ratings were carried out on the telephone by a trained research assistant who was blind to the treatment.

Observer rated mood rating scales were conducted by psychiatrists and consisted of the Montgomery Asberg Depression Rating Scale (MADRS) [18] and the Young Mania Rating Scale (YMRS) [19]. To define an “episode” based on the MADRS scores we used the recently defined categories for bipolar depression on the MADRS [21]. This study defined a MADRS score 0–6 as “normal”, 6–12 as “borderline”, 13–23 as “mild” and “moderate”, and >23 as “marked” and above. See Figure 1 for timing of LIFE, MADRS, and YMRS administration points.

### 2.4. Primary Outcomes

In Study 1, the primary outcome was the cumulative burden of depressive symptoms as measured by the LIFE. Study 2 had two primary outcomes: time to relapse and readmission, also using the LIFE.

### 2.5. Statistical Analysis

Statistical analyses were conducted using the Statistical Package for Social Sciences (SPSS) version 26 for Windows, IBM Corp, Chicago, IL, USA. The primary analysis was a univariate ANOVA completed separately for depression scores and mania scores. For depression scores, MADRS score (for the week corresponding to the week the MADRS was undertaken) was the dependent variable, LIFE score at the same time-point was a co-variate, time was a fixed factor, and patient was a random factor. The same analysis was conducted for the measures of mania using the YMRS. A simple bivariate correlation was also conducted to produce an overall correlation value.

A further univariate ANOVA was conducted to examine whether the relationship between LIFE score and MADRS score was moderated by diagnosis (bipolar I vs bipolar II) or MADRS score at the time the LIFE was conducted, with these factors added as variables. Because of a very low number of higher YMRS scores, the same analysis was not completed for mania ratings.

To examine the concordance of classification of “episode”, we first calculated the mean MADRS and YMRS scores which corresponded to each level on the LIFE for both depression and mania. Secondly, we examined the specificity and sensitivity of cut-offs on the LIFE to determine a MADRS score of >12 (“mild” or above according to Thase et al. [21]) using receiver operating characteristic (ROC) curves. Because of very few scores greater than 8 (usual cut off for relapse) on YMRS, we did not conduct a similar analysis for mania.

### 2.6. Ethics

Ethical approval was granted by the Canterbury Ethics Committee (Study 1) and the New Zealand Health and Disability Committee (Study 2). Registration was with the Australia and New Zealand Clinical Trials Registry (Study 1—ACTRN12605000722695; Study 2—ACTRN12611000961943).

## 3. Results

### 3.1. Characteristics of Total Sample

The analysis included data from 137 patients over 18 months across two studies. Clinical characteristics are shown in Table 1. Mean MADRS score and mean YMRS score for each rating on the LIFE are shown in Table 2. Of a possible 1370 observations (10 × 137 patients), a data point was missing either for MADRS or LIFE depression or YMRS or LIFE mania pairs on 138 occasions, leaving 1232 comparison points.

### 3.2. Depression Symptoms

In the univariate ANOVA for depression symptoms, there was a significant correlation between MADRS ratings and LIFE ratings (df 1,1045, F = 202.6, *p* < 0.001). There was also a significant interaction between time and LIFE (df 9,1035, F = 2.7, *p* = 0.004), indicating that the effect of the timing of the observation on the correlation between MADRS and LIFE mania was significant. The unadjusted bivariate correlation between all MADRS and LIFE depression measures was r = 0.57

An examination of the moderating effect of diagnosis and mood at the time of LIFE rating showed no significant effect of diagnosis (df 1,996, F = 1.25, *p* = 0.26). There was a significant effect of MADRS at the point of administration of LIFE (df 34,996, F = 3.11, *p* < 0.001). A post hoc calculation of correlations showed that this was higher where participants scored 8 or less on MADRS when the LIFE was done (n = 617, r = 0.57) than when participants scored >8 when LIFE was done (n = 297, r = 0.45).

### 3.3. Mania Symptoms

In the univariate ANOVA, for mania symptoms, there was a significant correlation between YMRS ratings and LIFE mania ratings (df 1,1045, F = 95.6, *p* < 0.001) and a significant interaction between time and LIFE mania (F = 3.5, *p* < 0.001), indicating that the effect of timing of the observation on the correlation between YMRS and LIFE mania was significant. Graphs of the correlations at each separate time-point for depression symptoms and mania symptoms are shown in Figure 2. The unadjusted bivariate correlation between all YMRS and LIFE mania measures was r = 0.40.

A ROC curve (see Figure 3) showed that for “mildly depressed”, determined using a cut off of >12 on the MADRS, area under curve = 0.78. Optimizing Youden’s statistic indicated that a cut off of >2 on the LIFE depression scale gave 70% sensitivity and 80% specificity in determining “mild depression”.

## 4. Discussion

This analysis suggests that overall, there is a moderately good correlation between retrospective assessment of mood state on the LIFE, and clinician rated mood rating scales carried out at the equivalent time point. The correlation was greater for assessment of depression symptoms compared with mania symptoms. The two sets of ratings correlated more closely at the beginning and the end of the period of assessment for the LIFE, at least for ratings of depressive symptoms. Using a cut-off on the MADRS to determine how well LIFE scores predicted “mildly depressed” [21] as defined by the MADRS, there was 70% sensitivity and 80% specificity for a LIFE depression score of 3. The data is of interest since the LIFE is a very low burden way of assessing long-term mood symptoms. Our raters generally take only 20 to 30 minutes to complete each LIFE interview.

That the ratings correlate relatively well suggests that on average people with BD are reasonably accurate at giving an account of mood symptoms over the previous 6 months. To our knowledge, there is only one previous study investigating the validity of a similar retrospective LIFE charting method in comparison with concurrent mood ratings. Albers (2015 [22]) examined a slightly different measure—the Life Chart [23] in 285 elderly (>60 years) patients with depression (see also [24]). The Life chart was conducted 6-monthly over 2 years and for each month rated depression as “no depression”, “depression but not chronic” and “chronic depression”. Therefore, the categorization was not as detailed as that used in the LIFE in the current study. Mood symptoms were also measured each 6 months using the Inventory of Depressive Symptoms (IDS) questionnaire [25]. A high percentage of ratings on the IDS classified the patients as severely depressed (12–25%). The correlation between LIFE Chart Burden and IDS scores ranged from 0.19 at baseline to 0.45 at 2-year follow-up. These correlations are less than in our study; however, patients were elderly and usually more depressed. The authors also found that the correlation was significantly influenced by the level of depression at the time the interview was conducted. When patients were more depressed, their retrospective mood ratings accorded less well with concurrent ratings. This is also in keeping with our results which found a significant moderating effect of depression symptoms at the time of LIFE interview on the correlation between LIFE ratings and MADRS ratings. When participants were more depressed when doing the LIFE interview, correlations with MADRs were lower, although there was still a significant correlation. This suggests, though, that patients do view the previous 6 months more negatively when under the influence of being currently depressed.

Another factor which may lessen concordance between ratings is cognitive function at the time of the LIFE rating. Albers et al. [22] found that worsening cognitive function was associated with less concordance between ratings. Significant cognitive impairment was of course much more likely in Albers’ study since this was in the elderly. In our study, in contrast to Albers et al., the patients had BD and they were much younger. In BD, there are significant differences in mean cognitive scores compared with healthy people (see [26] for review), and depending on mood state and severity, a percentage of patients will have significant cognitive impairment [27]. In one of the studies examined here we did measure cognitive function [28], finding a modest improvement over 18 months of treatment. However, we had repeat testing in only 78 participants at 18 months (there was no testing at 6 or 12 months), leaving relatively few time points to analyze the effects of cognition on correlations. We did not therefore undertake this analysis.

Many other studies have compared self-report questionnaire ratings with clinician-administered interview-based mood ratings. However, these have been conducted concurrently. They are difficult to interpret because of the very large variation in correlation between measures. For example, Richter et al. [29] reviewed correlation between the Beck Depression Inventory (BDI) and clinician-rated scales, finding correlations ranging from 0.19 to 0.73. They argued that studies examining severely depressed in-patients may give a particularly poor correlation, in keeping with our finding of less correlation when patients were more depressed. The overall correlation of 0.57 seen here for depression ratings is better than several of the studies reviewed, despite, of course, the rating used here being retrospective.

Of course, the two measures do not correspond perfectly. There are several potential issues in the LIFE interview process which may contribute to this. First, patients may have a poor recall of when in that 6 months their mood was disturbed. This issue may account for some of the loss of correlation (i.e., patients may have a relatively good recall of overall mood disturbance but not the exact timing of disturbance, resulting in the correlations with observer ratings at an exact time point being poor). The modifying effect of time in this regard is significant, and inspection of Figure 2 suggests that correlations are poorer for ratings in the middle of the 6-month time period for depression ratings—but although there is a significant moderating effect of time on mania ratings, the pattern is not clear. This is unsurprising for ratings of weeks at the end of the 6-month period since these are more recent. The better correlations at the beginning of the 6-month period for depression ratings may relate to this being more easily anchored in time. Raters go through a process of first identifying key events in the 6-month period as anchor points and then discussing mood around these anchor points. The anchor points immediately after the last interview may serve as universal points which anchor memory for most participants. Studies have used a LIFE conducted at three monthly intervals [15]. Both logic and our results suggest that concordance with concurrent ratings will be higher when the LIFE is used in this way. Second, poor correlation may indicate that symptoms are relatively mild and therefore variations at the lower end of the scale are difficult to remember and rate accurately. It may be for this reason that the correlation for mania scores was less strong since most mania ratings were low. The other possible issue in the rating of mania is that lack of insight may prevent the retrospective reporting of symptoms. However, given the relatively few episodes of mania and the low level of symptoms, we do not believe that this analysis can examine this issue.

Using a categorical definition of ‘in-episode’ or not on the MADRS allowed us to estimate how specific and sensitive the LIFE is in determining when a patient is, using the MADRS, defined as having significant symptoms and is in effect “relapsed”. To determine this, we used a cut-off on the MADRS of 13—“mild depression” [21]. The cut off of >2 on the LIFE gave reasonable sensitivity (70%) and specificity (80%). In effect, if a cut-off of 2 on the LIFE was used to define a state of “mild depression”, this would miss 30% of cases but only classify 20% of those who do not meet this criterion on MADRS as “mildly depressed”.

### 4.1. Limitations and Future Directions

The study has several limitations. First, it was not originally designed for this purpose and the data is pooled from two separate studies. This then represents a secondary analysis of data in which the concurrent ratings were done by chance rather than by design. Second, mania was rare, and scores on the YMRS were low, making data regarding mania less robust and not necessarily applicable to more severe and frequent episodes of mania. Third, we were not able to address the issue of the effect of cognitive impairment on mood ratings in this analysis. Fourth, an important question is whether in fact the score on a clinician-administered mood scale is the best measure of mood symptoms. It could be argued that either patients’ own ratings on a mood rating scale, or indeed their own perception of their mood over the previous 6 months, are outcomes which are more important to patients. In this study we did not have concurrent self-rated scales. Had these been available we would have hypothesised a closer correlation than with the clinicians rated scales—at least for depression. Last, a broader question which may be explored further in future research is how measures of mood symptom burden correlate with measures of various aspects of functioning or with measures which are known to be of importance for people with BD. As noted, scales have been developed by people with lived experience of BD which may capture what is important for people living with the illness [30]. Future research should examine the relationship between symptom burden and quality of life measured in this way.

### 4.2. Conclusions

Despite the limitations noted above, the study is able to compare ratings made retrospectively using the LIFE with those made at the same time-point, concurrently, using clinician-administered mood ratings scales, and found that there was surprisingly good correlation. This is important given that the LIFE is a very efficient way of measuring mood symptom burden. We note that it is likely to be more accurate if conducted more frequently, but this then becomes a trade-off between more burden and greater accuracy. We note that in future there could be further examination regarding how these measures correlate with measures developed by those with lived experience.

## Figures and Tables

**Figure 1 brainsci-12-01717-f001:**
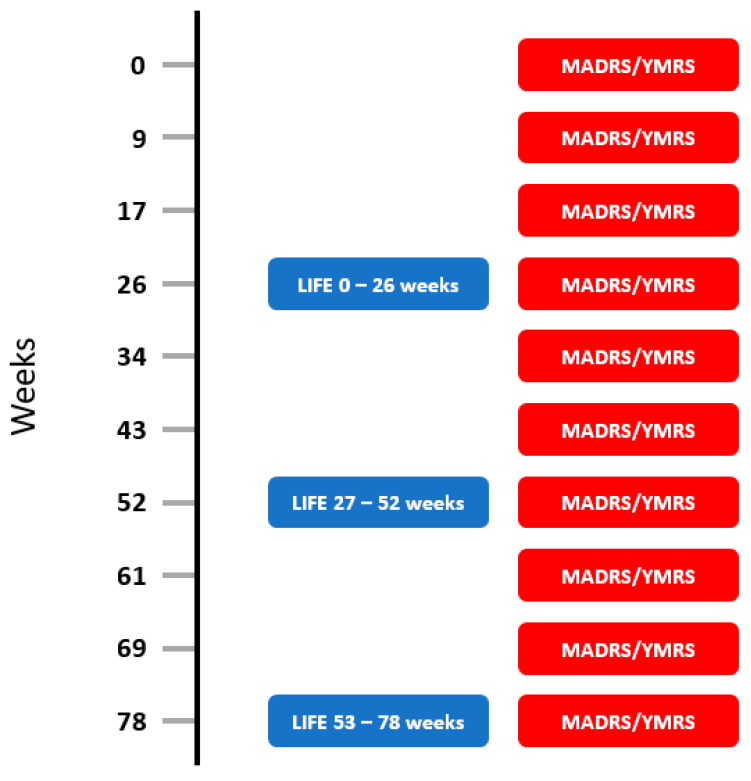
Timing of Longitudinal Interval Follow-up Evaluation (LIFE) and mood ratings using the Montgomery-Asberg Depression Rating Scale (MADRS) and the Young Mania Rating Scale (YMRS).

**Figure 2 brainsci-12-01717-f002:**
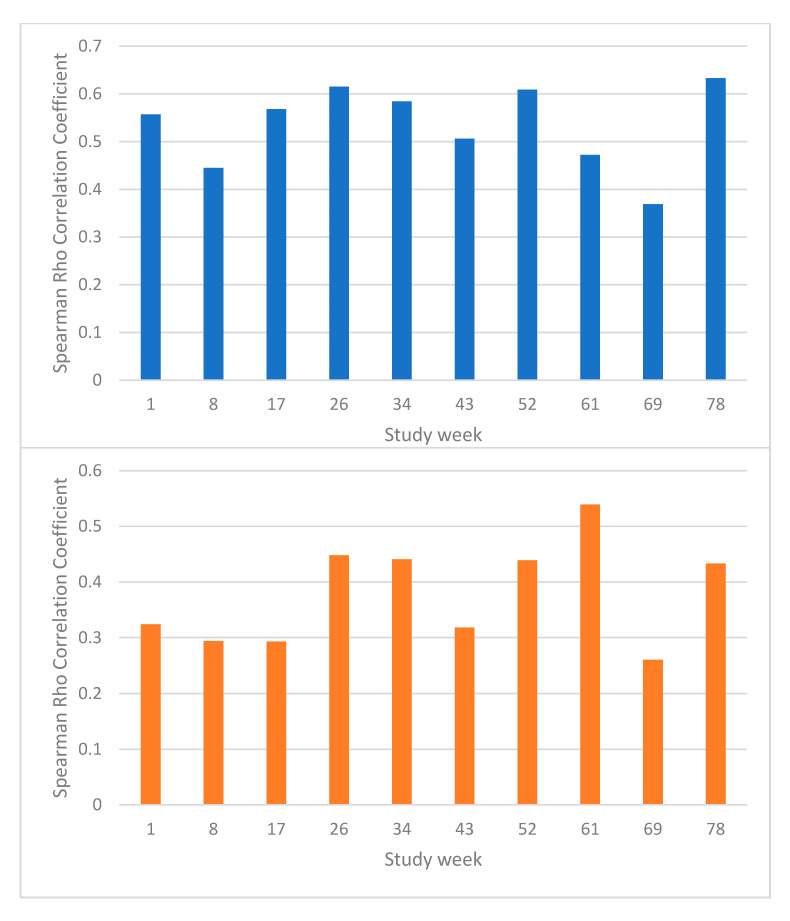
Correlations between Longitudinal Interval Follow-up Evaluation (LIFE) ratings and mood rating scales at different time-points—top panel LIFE depression vs Montgomery-Asberg Depression Rating Scale (MADRS), bottom panel LIFE mania vs Young Mania Rating Scale (YMRS).

**Figure 3 brainsci-12-01717-f003:**
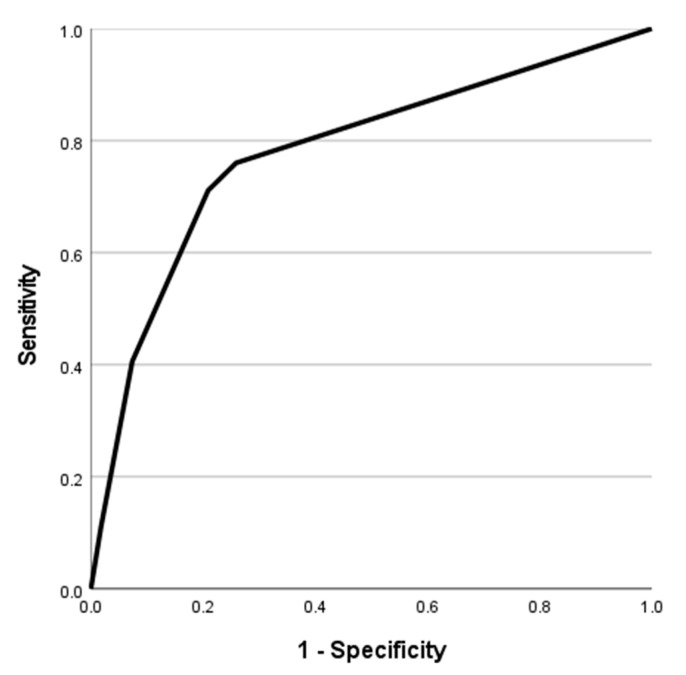
Receiver operating characteristics (ROC) curve of Longitudinal Interval Follow-up Evaluation (LIFE) depression score—sensitivity and specificity in determining “mildly depressed” status (defined by Montgomery-Asberg Depression Rating Scale [MADRS] score > 12).

**Table 1 brainsci-12-01717-t001:** Clinical Characteristics of Study Groups.

	Study 1		Study 2	
	(N = 95)		(N = 42)	
Characteristic	N	%	N	%
Age (M ± SD)	26.4 ± 5.9 **		39.6 ± 14.5 **	
Gender (% female)	72	76	32	76
Ethnicity (% Pākehā)	78	82	32	76
Bipolar I (%)	78 *	82	28 *	67
Index episode (% depression)	87*	92	32*	76
Rapid cycling (%)	31	33	7	17
Age at onset (M ± SD)	15.0 ± 5.1 *		17.4 ± 6.9 *	
Medication use ^†^				
Lithium	28	29	12	29
Anticonvulsant mood stabiliser	36	38	17	41
Antipsychotic	49	52	20	49
Antidepressant	49	52	22	54
SAS total score ^†^ (M ± SD)	2.3 ± 0.5		2.1 ± 0.5	
Cumulative mood score (LIFE) ^‡^ (M ± SD)	0.9 ± 0.6 **		0.5 ± 0.6 **	

^†^ = At week 0; ^‡^ = At week 26, reflective of the cumulative mood score over the first 26 weeks; * = Significantly different at the *p* < 0.05 level. ** = Significantly different at the *p* < 0.01 level.

**Table 2 brainsci-12-01717-t002:** Mean Mood Rating Scale Scores at Each Longitudinal Interval Follow-up Evaluation (LIFE) Score.

LIFE Depression Score (no. of Observations)	MADRS Mean (SD)	LIFE Mania Score (no. of Observations)	YMRS Mean (SD)
1 (720)	4.6 (6.5)	1 (994)	1.2 (2.7)
2 (59)	9.2 (7.7)	2 (45)	3.6 (4.1)
3 (209)	12.9 (8.7)	3 (96)	4.4 (5.4)
4 (144)	17.7 (9.9)	4 (43)	6.3 (6.0)
5 (53)	20.4 911.9)	5 (6)	11 (11.5)
6 (2)	26.5 (16.3)	6 (3)	4.7 (3.7)

## Data Availability

The data presented in this study are available on request from the corresponding author. The data are not publicly available due to ethical reasons.

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
