# Peer review of "Validation of the Longitudinal Interval Follow-Up Evaluation for the Long-Term Measurement of Mood Symptoms in Bipolar Disorder"

_brainsci, 2022, doi:10.3390/brainsci12121717_

Round 1

Reviewer 1 Report

The present research article by Porter and colleagues, entitled ‘Long-term measurement of mood symptoms in Bipolar Disorder: Relationship between scores on the Longitudinal Interval Follow-up Evaluation with concurrent measures on clinician mood rating scales’ is a well-written and useful summary on the status of knowledge of the effectiveness of LIFE retrospective scale in assessing mood symptoms. For this purpose, authors selected data from two clinical trials of 18 months of psychotherapy for patients with bipolar depression were analysed, comparing data from the Montgomery-Asberg Depression Rating Scale (MADRS), the Young Mania Rating Scale (YMRS) and LIFE, for 8-9 weeks and then after each 6 months. Results showed significant correlations between LIFE depression ratings and concurrent MADRS score, and between LIFE mania ratings and YMRS score. Authors concluded by stating correlations scores were higher for depression scores, and were high even when the LIFE rating was several months before the interview, suggesting that the LIFE has validity in examining burden of mood symptoms over time, with relatively little burden of assessment.

The main strength of this manuscript is that it addresses an interesting and timely question, describing how LIFE retrospective interview is able to compare mood symptoms burden, just like clinician-administered mood rating scales. In general, I think the idea of this article is really interesting and the authors’ fascinating observations on this timely topic may be of interest to the readers of Brain Sciences. However, some comments, as well as some crucial evidence that should be included to support the author’s argumentation, needed to be addressed to improve the quality of the manuscript, its adequacy, and its readability prior in the present form, in particular reshaping parts of the Introduction and Methods sections by adding more evidence and theoretical constructs.

Please consider the following comments:

·       I suggest changing the title. In my opinion, in the present form it seems to be too wordy and not enough clear and specific.

·       Abstract: According to the Journal’s guidelines, the abstract should be a total of about 200 words maximum. Also, according to the Journal’s guidelines, this section should be presented as a single paragraph, without explicit sub-headings. Please correct the actual one.

·       In general, I recommend authors to use more evidence to back their claims, especially in the Introduction of the article, which I believe is currently lacking. Thus, I recommend the authors to attempt to deepen the subject of their manuscript, as the bibliography is too concise: nonetheless, in my opinion, less than 50 articles for a research article are really insufficient. Therefore, I suggest the authors to focus their efforts on researching more relevant literature: I believe that adding more studies and reviews will help them to provide better and more accurate background to this study.

·       Introduction: The ‘Introduction’ section is well-written and nicely presented, with a good balance of descriptive text and information about etiology and characteristics of mood disorders, specifically bipolar disorder. Nevertheless, I believe that more information about neuro-pathophysiology and core features of these disorders will provide a better and more accurate background, because as it stands, this information is not highlighted in the text. In this regard, I would suggest to add more information on pathological neural substrates of mood disorders, like depression or anxiety, specifically on structural as well as functional abnormalities of specific brain regions (i.e., prefrontal cortex), and on related effects on patients’ cognitive impairments. In my opinion, authors could further explore significant functional brain alterations and impaired brain circuits in mood disorders (https://doi.org/10.1016/j.tins.2022.04.003https://doi.org/10.1111/psyp.14122), and focus on relationship between the molecular regulation of higher-order neural circuits and neuropathological alterations in these diseases (https://doi.org/10.3390/cells11162607https://doi.org/10.3390/biomedicines9050517).

·       Measures: Data about participants and information about clinical assessment for patients’ selection are not adequately explained, and cannot be listed down only in summary tables. For this reason, I would ask the authors to provide more information about the diagnostic tests used for clinical evaluation.

·       Statistical analysis: Please specify the acronym ‘SPSS’ next to the full name ‘Statistical Package for the Social Sciences’. Also, I was wondering why the Authors did not perform also more robust statistical analysis (i.e., ANOVAs) to compare depressive and mania symptoms.

·       Results: In my opinion, this section is well organized, but it illustrates findings in an excessively broad way, without really providing full statistical details, to ensure in-depth understanding and replicability of the findings. I suggest rewriting this section more accurately, and to present statistical data not only in the main text, but also in tables.

·       In my opinion, I think that a proper and defined ‘Conclusions’ paragraph would be very useful to state some thoughtful as well as in-depth considerations by the authors. In this section, Authors should make an effort, trying to explain the theoretical implication as well as the translational application of their research.

·       In according to the previous comment, I would ask the authors to include a proper ‘Limitations and future directions’ section before the end of the manuscript, in which authors can describe in detail and report all the technical issues brought to the surface.

·       Tables: According to the Journal’s guidelines, please provide a short explanatory caption for the table within the text.

·       References: Authors should consider revising the bibliography, as there are several incorrect citations. Indeed, according to the Journal’s guidelines, they should provide the abbreviated journal name in italics, the year of publication in bold, the volume number in italics for all the references.

Overall, the manuscript contains 4 figures, 2 tables and 25 references. I believe that the manuscript might carry important value in describing how LIFE retrospective interview is able to compare mood symptoms burden, just like clinician-administered mood rating scales.

I hope that, after these careful revisions, this paper can meet the Journal’s high standards for publication.

I am available for a new round of revision of this paper. I declare no conflict of interest regarding this manuscript.

Best regards,

Reviewer

Author Response

The present research article by Porter and colleagues, entitled ‘Long-term measurement of mood symptoms in Bipolar Disorder: Relationship between scores on the Longitudinal Interval Follow-up Evaluation with concurrent measures on clinician mood rating scales’ is a well-written and useful summary on the status of knowledge of the effectiveness of LIFE retrospective scale in assessing mood symptoms. For this purpose, authors selected data from two clinical trials of 18 months of psychotherapy for patients with bipolar depression were analysed, comparing data from the Montgomery-Asberg Depression Rating Scale (MADRS), the Young Mania Rating Scale (YMRS) and LIFE, for 8-9 weeks and then after each 6 months. Results showed significant correlations between LIFE depression ratings and concurrent MADRS score, and between LIFE mania ratings and YMRS score. Authors concluded by stating correlations scores were higher for depression scores, and were high even when the LIFE rating was several months before the interview, suggesting that the LIFE has validity in examining burden of mood symptoms over time, with relatively little burden of assessment.

The main strength of this manuscript is that it addresses an interesting and timely question, describing how LIFE retrospective interview is able to compare mood symptoms burden, just like clinician-administered mood rating scales. In general, I think the idea of this article is really interesting and the authors’ fascinating observations on this timely topic may be of interest to the readers of Brain Sciences. However, some comments, as well as some crucial evidence that should be included to support the author’s argumentation, needed to be addressed to improve the quality of the manuscript, its adequacy, and its readability prior in the present form, in particular reshaping parts of the Introduction and Methods sections by adding more evidence and theoretical constructs.

Please consider the following comments:

I suggest changing the title. In my opinion, in the present form it seems to be too wordy and not enough clear and specific.

We have shortened the Title as suggested, which now reads: Validation of the Longitudinal Interval Follow-up Evaluation for the long-term measurement of mood symptoms in Bipolar Disorder

Abstract: According to the Journal’s guidelines, the abstract should be a total of about 200 words maximum. Also, according to the Journal’s guidelines, this section should be presented as a single paragraph, without explicit sub-headings. Please correct the actual one.

We have changed the abstract as requested and it now has no subheadings. We added a sentence at the end regarding future directions of this project, so it is now 215 words. However, if there is a preference for being under the word limit, we can delete this final sentence.

In general, I recommend authors to use more evidence to back their claims, especially in the Introduction of the article, which I believe is currently lacking. Thus, I recommend the authors to attempt to deepen the subject of their manuscript, as the bibliography is too concise: nonetheless, in my opinion, less than 50 articles for a research article are really insufficient. Therefore, I suggest the authors to focus their efforts on researching more relevant literature: I believe that adding more studies and reviews will help them to provide better and more accurate background to this study.

We believe that it is important to concisely cover literature that is relevant to this paper. However, as per the reviewer’s suggestion, we have added more detail and further references to the introduction, in particular related to EMA – a possible alternative method of measuring mood.

Introduction: The ‘Introduction’ section is well-written and nicely presented, with a good balance of descriptive text and information about etiology and characteristics of mood disorders, specifically bipolar disorder. Nevertheless, I believe that more information about neuro-pathophysiology and core features of these disorders will provide a better and more accurate background, because as it stands, this information is not highlighted in the text. In this regard, I would suggest to add more information on pathological neural substrates of mood disorders, like depression or anxiety, specifically on structural as well as functional abnormalities of specific brain regions (i.e., prefrontal cortex), and on related effects on patients’ cognitive impairments. In my opinion, authors could further explore significant functional brain alterations and impaired brain circuits in mood disorders (https://doi.org/10.1016/j.tins.2022.04.003; https://doi.org/10.1111/psyp.14122), and focus on relationship between the molecular regulation of higher-order neural circuits and neuropathological alterations in these diseases (https://doi.org/10.3390/cells11162607; https://doi.org/10.3390/biomedicines9050517).

We appreciate the reviewer’s comments but respectfully do not feel that discussion of neurobiology is relevant to this paper.

Measures: Data about participants and information about clinical assessment for patients’ selection are not adequately explained, and cannot be listed down only in summary tables. For this reason, I would ask the authors to provide more information about the diagnostic tests used for clinical evaluation.

As also per Reviewer 4, we have clarified the diagnostic interview used and added details of the rating scales and their administration to the Methods section (pg. 4). We have also expanded on the sections describing participants for both studies, including where they were recruited from (pg. 3).

Statistical analysis: Please specify the acronym ‘SPSS’ next to the full name ‘Statistical Package for the Social Sciences’. Also, I was wondering why the Authors did not perform also more robust statistical analysis (i.e., ANOVAs) to compare depressive and mania symptoms.

We have added the SPSS acronym. We did not conduct statistical analysis of differences in symptoms between the different studies because they are different studies done at different times and there was therefore no reason to conduct statistical comparisons.

Results: In my opinion, this section is well organized, but it illustrates findings in an excessively broad way, without really providing full statistical details, to ensure in-depth understanding and replicability of the findings. I suggest rewriting this section more accurately, and to present statistical data not only in the main text, but also in tables.

We are not clear what the reviewer specifically intends. The Results are presented in Tables and in the Figures – so we believe that the Figures are better representations of the Results than Tables would have been.

In my opinion, I think that a proper and defined ‘Conclusions’ paragraph would be very useful to state some thoughtful as well as in-depth considerations by the authors. In this section, Authors should make an effort, trying to explain the theoretical implication as well as the translational application of their research.

We agree with the reviewer, and have adapted the final paragraph so that it is a clearer Conclusions paragraph with some further comment on implications.

In according to the previous comment, I would ask the authors to include a proper ‘Limitations and future directions’ section before the end of the manuscript, in which authors can describe in detail and report all the technical issues brought to the surface.

We have done this, with this comment also having been suggested by Reviewer 3. The separate Limitations and future directions paragraph is now section 4.1 (pg. 11).

Tables: According to the Journal’s guidelines, please provide a short explanatory caption for the table within the text.

There is an explanatory caption but we do note that in Table 1 this would read better as “Clinical Characteristics of Study Groups” – thus avoiding the misconception that the 2 groups should be compared statistically. For Table 2 we believe that the current caption “Mean Mood Rating Scale Scores at Each Longitudinal Interval Follow-up Evaluation (LIFE) Score” is adequate.

References: Authors should consider revising the bibliography, as there are several incorrect citations. Indeed, according to the Journal’s guidelines, they should provide the abbreviated journal name in italics, the year of publication in bold, the volume number in italics for all the references.

We have corrected incorrectly formatted references.

Overall, the manuscript contains 4 figures, 2 tables and 25 references. I believe that the manuscript might carry important value in describing how LIFE retrospective interview is able to compare mood symptoms burden, just like clinician-administered mood rating scales

We thank the reviewer for these positive comments.

Reviewer 2 Report

No extra recommendation.

Author Response

We thank Reviewer 2 for this very positive review of our paper. No response is required.

Reviewer 3 Report

This is a very interesting paper focused on the correlations between scores in the Longitudinal Interval Follow-up assessment and assessment scales. The paper is well-written and is of interest for the journal. However, several minor changes should be made.

Abstract.

1- I recommend to explain the study design on the abstract section. Is it a correlational analyses? Please, provide more details.

2-Conclusions should be more directive and clear. Is there a good/strong correlation between both measures? What about future recommendations?

Introduction

1- The introduction section is brief. I recommend to expand the first paragraph by explaining the definition of bipolar disorder; which kind of episodes there are, and what about using LIFE evaluation for the identification of hypomanic episodes.

2- The main aims of the study should be explained separately, in a specific subsection.

Methods

1- At the beginning of this section, I recommend to describe the participants and the study design. The ethics subsection is preferred to be described later.

2- In the subsection of "Statistical analysis", the authors are describing how are they analyzing the episodes and the scores of the assessment scales. These should be previously mentioned, not only in the statistical analysis subsection.

Results

1-Table 1. Bipolar 1 is not correct. Please change it to Bipolar I.

2-The results section should be divided into several subsections. I would be better to classify results into: Main characteristics of the total sample.

3- The correlational analyses were described for depressive and mania symptoms. Please, describe more details from the analyses and divide these into two subsections.

Discussion

1- The discussion section needs a limitation and strenghts subsection. The last paragraphs have been focused on this topic.

2- A conclusions section is needed. Please explain future perspectives for planned studies.

Author Response

This is a very interesting paper focused on the correlations between scores in the Longitudinal Interval Follow-up assessment and assessment scales. The paper is well-written and is of interest for the journal. However, several minor changes should be made.

Abstract.

I recommend to explain the study design on the abstract section. Is it a correlational analyses? Please, provide more details.

We have made it clearer in the Abstract that the primary analysis was correlational.

2-Conclusions should be more directive and clear. Is there a good/strong correlation between both measures? What about future recommendations?

We struggled with a definitive statement regarding what is a “good/strong” correlation and have noted only that the correlations seen were, according to our hypothesis “surprisingly good”. We have added the recommendation that in future studies should examine the relationship between mood symptom burden measured in this way and patient derived measures of well-being.

Introduction

The introduction section is brief. I recommend to expand the first paragraph by explaining the definition of bipolar disorder; which kind of episodes there are, and what about using LIFE evaluation for the identification of hypomanic episodes.

We have added a definition of mood disorders (MDD and BD) to the first paragraph of the Introduction, and have noted the use of the LIFE to determine “relapse” and overall mood burden now in paragraph 3 of the Introduction.

The main aims of the study should be explained separately, in a specific subsection.

The aims of the analysis are now provided, in a separate section (section 1.1 Aims) at the end of the Introduction.

Methods

At the beginning of this section, I recommend to describe the participants and the study design. The ethics subsection is preferred to be described later.

Thank you – we have made this change, with the Ethics section now following the ‘Statistical Analysis section, at the end of the Methods.

In the subsection of "Statistical analysis", the authors are describing how are they analyzing the episodes and the scores of the assessment scales. These should be previously mentioned, not only in the statistical analysis subsection.

We agree that this is perhaps not the right section for this information. We have shifted description of how episodes were defined using the MADRS to the Measures section of the Methods (currently section 2.4).

Results

1-Table 1. Bipolar 1 is not correct. Please change it to Bipolar I.

Done.

The results section should be divided into several subsections. I would be better to classify results into: Main characteristics of the total sample. The correlational analyses were described for depressive and mania symptoms. Please, describe more details from the analyses and divide these into two subsections.

In relation to these comments, we have separated the Results section into three main sections: 3.1 Characteristics of the total sample, 3.2 Depression symptoms, and 3.3 Mania symptoms, as suggested. We have also provided more details on the analyses described in the Depression and Mania sections (sections 3.2 and 3.3).

Discussion

The discussion section needs a limitation and strengths subsection. The last paragraphs have been focused on this topic.

We have made this a clearly separate section with a heading as suggested also by Reviewer 1, which is now on pg. 11 of the manuscript.

A conclusions section is needed. Please explain future perspectives for planned studies.

Once again this is in accord with Reviewer 1 – we thank both reviewers and have made a more clearly focused Conclusions section (section 4.2, pg. 12).  Future directions are mainly in the previous section of the Discussion (section 4.1), as we felt they flowed on from the limitations nicely.

Reviewer 4 Report

Dear Editor,
I really appreciate the opportunity to review the manuscript brainsci-2051949 entitled:
"Long-term measurement of mood symptoms in Bipolar Disorder: Relationship between scores on the Longitudinal Interval Follow-up Evaluation with concurrent measures on clinician mood rating scales"

I commend the authors for describing this critical and timely issue. The paper is interesting and well-written; however, I would like to highlight some issues that merit revision:

on page 6 it is reported that SCID-I was used for DSM-IV (1994), I guess the authors meant DSM-IV-TR (2000), but since it is very old anyway, DSM-5 having come out in 2013, it should be corrected and possibly reported among the limitations the use of the manual and its outdated SCID.

Author Response

Thank you for this important comment about the version of the SCID-I used in the studies reported in this paper. 

The version we used was indeed the SCID-I DSM-IV (not DSM-IV-TR). We have provided the correct reference as a comment in the revised paper, as we have realised when considering this comment that we had referenced the incorrect version of the SCID in this paper, and we apologise for this.

The DSM-5 had not been published when the first study was conducted or at the point when the second study had commenced recruitment. We also note that the SCID-I for DSM-IV is designed to yield psychiatric diagnoses consistent with DSM-IV/DSM-IV-TR diagnostic criteria. DSM-IV-TR was purely a text revision, rather than having made any changes to diagnostic criteria (see First & Pincus, 2002, https://doi.org/10.1176/appi.ps.53.3.288, for a description of this). We therefore do not believe that including this as a Limitation in this paper is warranted.

Round 2

Reviewer 1 Report

The authors did an excellent job clarifying all the questions I have raised in my previous round of review. Currently, this paper entitled ‘Long-term measurement of mood symptoms in bipolar disorder: Relationship between scores on the Longitudinal Interval Follow-up Evaluation with concurrent measures on clinician mood rating scales’, is a well-written, timely piece of research that described the effectiveness of LIFE retrospective scale in assessing mood symptoms.

Overall, this is a timely and needed work. It is well researched and nicely written, therefore I believe that this paper does not need a further revision, therefore the manuscript meets the Journal’s high standards for publication.

I am always available for other reviews of such interesting and important articles.

Thank You for your work.